# Variation in salivary cortisol responses in yearling Thoroughbred racehorses during their first year of training

**Amy R. Holtby**[1,2]*, **Beatrice A. McGivney**[2], **John A. Browne**[1], **Lisa M. Katz**[3], **Keith J. Murphy**[4], **Emmeline W. Hill**[1,2]

**1** UCD School of Agriculture and Food Science, University College Dublin, Belfield, Dublin, Ireland, **2** Plusvital Ltd., The Highline, Dun Laoghaire, Co. Dublin, Ireland, **3** UCD School of Veterinary Medicine, University College Dublin, Belfield, Dublin, Ireland, **4** UCD Neurotherapeutics Research Group, Conway Institute, UCD School of Biomolecular and Biomedical Science, University College Dublin, Belfield, Dublin, Ireland

* amy.holtby@plusvital.com

## Abstract

Thoroughbred horses are bred for competitive racing and undergo intense training regimes. The maintenance of physical soundness and desirable behavioural characteristics are critical to the longevity of a racing career. Horses intended for Flat racing generally enter training as yearlings and undergo introductory training prior to exercise conditioning for racing. This period requires rapid adjustment to a novel environment. As a prey animal, a horse's 'fight-or-flight' response is highly adapted, in which a well-understood component of this response, the hypothalamic-pituitary-axis, is activated in response to a stress stimulus, releasing cortisol. In the Thoroughbred, a significant difference in salivary cortisol concentrations between pre- and post-first time ridden (i.e., first backing) by a jockey have previously been identified. Here, to test the hypothesis that salivary cortisol concentrations may be used to objectively detect individual variations in the acute physiological stress response we investigate individual variation in cortisol response to training milestones. Saliva samples were collected from a cohort of $n = 96$ yearling Flat racehorses, at the same training yard, across three timepoints at rest: before entering the training yard ($n = 66$), within three days of entry to the training yard ($n = 67$) and following 2–3 weeks in the training yard ($n = 50$). Salivary cortisol concentration was measured using an ELISA. There was no significant difference in cortisol concentration (ANOVA, $P > 0.05$) across the samples collected at timepoints at rest. Samples were also collected before and 30 minutes after exposure to three novel training events: first time long-reined ($n = 6$), first time backed by a jockey ($n = 34$), and first time ridden on the gallops ($n = 10$). Mean salivary cortisol concentration after all three novel training events was significantly higher than prior to the training event (Paired t-test, $P < 0.005$). The ranges of post-event salivary cortisol concentration across all timepoints suggest individual variation in the measured stress response, reflecting individual differences in stress response to the early training environment. This measure may be used as an objective assessment of the stress response of Thoroughbred racehorses during training.

**Data Availability Statement:** All relevant data are within the paper and its Supporting Information files.

**Funding:** This research was carried out with the financial support of Plusvital Ltd. The funder provided support in the form of salaries for authors ARH, BAM and EWH, but did not have any additional role in the study design, data collection and analysis, decision to publish, or preparation of the manuscript. The specific roles of these authors are articulated in the 'author contributions' section.

**Competing interests:** This research was carried out with the financial support of Plusvital Ltd. Plusvital is an equine nutrition and genetic testing company in which EWH is a shareholder. This does not alter our adherence to PLOS ONE policies on sharing data and materials.

## Introduction

The Thoroughbred horse is a product of more than 300 years of selection for athletic traits [1] and is generally maintained in an environment vastly different to the wild habitats of ancestral horses. Evolutionary and domestication processes have shaped an expanded neocortex in the horse, required for processing of sensory inputs [2]. In the wild, hyperarousal, often described as the 'fight or flight' response, originated with the need to escape and survive environmental dangers, particularly predation [3, 4].

Psychological stress can elicit the same stress response observed in the 'fight or flight' response to a perceived threat, inducing neuroendocrine responses. This primarily involves activation of the hypothalamic-pituitary-adrenal (HPA) axis and sympathetic nervous system (SNS), leading to increased secretion of glucocorticoids and catecholamines [5]. Acute stress responses are normally evolved processes that are protective whereas chronic over-activity of the stress system increases the risk of disease [6]. Chronic stimulation of the acute stress response (i.e., chronic stress) has deleterious effects on the brain, particularly impacting the hippocampus, an area that regulates the stress response [7], leading to cognitive and mood disturbances [8]. Behavioural plasticity enables a horse to experience new situations with reduced stimulation of the acute stress response. Identifying horses with more sensitive temperaments, characterised by greater acute stress responses, may enable more proactive management strategies during this critical period of learning and development, ultimately reducing the long-term effects of stress.

Cortisol is the primary glucocorticoid produced in the adrenal cortex, with acute stress eliciting rapid increases in blood cortisol concentration. Only 5–10% of blood cortisol is biologically active and unbound to protein; this portion rapidly diffuses into saliva such that salivary cortisol concentration effectively mirrors blood cortisol concentration [9]. Cortisol levels are under circadian control with the highest levels detected in the early morning [10]. Salivary cortisol concentrations have been shown to correspond with serum cortisol levels [9], with many studies in horses now measuring salivary cortisol concentration to evaluate the acute stress response to handling and training procedures [11–13]. Collection of salivary samples is considered to be a less-invasive method for measurement of cortisol when compared with that of venepuncture, which itself may cause an acute stress response [14]. Using salivary cortisol measures, it has been previously reported that the first time a horse is mounted by a rider, also known as 'backing', is the most stressful event in the early training of Thoroughbred horses [15]. Salivary cortisol levels have also been shown in Thoroughbreds [22] and sport horses [23] to be significantly higher post-exercise compared to resting values.

Husbandry practices for competitive horses often result in exposure to an unnatural environment with restricted movement and social isolation, contrary to a horse's natural behaviour [16]. As well as during the early training period, it is common for racehorses to encounter novel situations and environments during their racing career, such as when travelling to and racing at unfamiliar locations. Trainers and handlers modify individual training regimes and the environment for each horse according to subjective observations of how they adjust to different events during the training regime to ensure high standards of welfare and productivity of the horses in their care. This perceived adjustment by the horse is often referred to as 'coping'.

Although racehorse training is recognised to be one of the more stressful types of equine training programmes due to the housing typically used (individual stalls *vs*. herds in a field) and exercise and performance requirements (high-intensity exercise training, frequent travel),

not all horses raised in the same environment experience the same degree of stress. Some horses develop an 'active coping style' characterised by behavioural hyperactivity (stall walking, crib-biting) and deficits in cognitive flexibility, both of which can reduce responses to effective training and add further stress to the horse [17]. Horses experiencing chronic stress, with the development of behavioural hyperactivity, may have problems maintaining a good body condition and are at increased risk of certain diseases [18, 19]. Horses not employing an active coping style may exhibit behaviours considered dangerous as part of a 'reactive' style [20]. Furthermore, horses that do not easily adapt to a high-pressure training environment

**Table 1. Overview of cohorts for total number of samples collected at each time point.**

| Timepoint | Cohort | Sample collection details |
|---|---|---|
| Timecourse 1(T1) | $n = 5$ | Samples collected from stabled horses at rest every two hours between 08:00 (T1) and 16:00 (T5) inclusive. |
| Timecourse 2 (T2) | | |
| Timecourse 3 (T3) | 5 female | |
| Timecourse 4 (T4) | | |
| Timecourse 5 (T5) | 0 male | |
| Resting Nursery (RN) | $n = 66$ | Samples collected from stabled horses at rest between 14:00–15:30 at the nursery yard prior to entering main training environment. |
| | 34 female | |
| | 32 male | |
| Resting Main Yard (RY_T1) | $n = 67$ | Samples collected from stabled horses at rest between 14:00–15:30 within 3 days of entering main yard training environment. |
| | 33 female | |
| | 34 male | |
| Follow-up resting Main Yard (RY_T2) | $n = 50$ | Samples collected from stabled horses at rest between 14:00–15:30 2–3 weeks after entering main yard training environment. |
| | 26 female | |
| | 24 male | |
| Pre-driving (FD_T1) | $n = 6$ | Samples collected from stabled horses at rest before tack applied. |
| | 5 female | |
| | 1 male | |
| Post-Driving (FD_T2) | $n = 6$ | Samples collected from stabled horses ~30 mins after tack applied and first time being 'driven' using long reins. |
| | 5 female | |
| | 1 male | |
| Pre-backing (FR_T1) | $n = 38^*$ | Samples collected from stabled horses at rest before tack applied. |
| | 14 female | |
| | 24 male | |
| | $n = 34$ paired samples | |
| Post-backing (FR_T2) | $n = 43^*$ | Samples collected from stabled horses ~30 mins after tack applied and first time being backed by jockey. |
| | 19 female | |
| | 24 male | |
| | $n = 34$ paired samples | |
| Pre-first time on gallops (FG_T1) | $n = 10$ | Samples collected from stabled horses at rest before tack applied. |
| | 2 female | |
| | 8 male | |
| Post-first time on gallops (FG_T2) | $n = 10$ | Samples collected from stabled horses ~30 mins after tack applied and being ridden on the gallops with a group of other yearling horses for the first time. |
| | 2 female | |
| | 8 male | |

*for Pre/post-backing samples n = 34 horses had both pre- and post-backing samples available. Instances in which there are un-paired samples are due to horses being unavailable for sampling.

may have extended periods out-of-training to allow them to mature/settle into the training schedule, resulting in a delay in racing and a negative impact on economic earnings. It has been reported in an Australian population of Thoroughbreds that 'unsuitable temperament/ behaviour' is responsible for >6% of Thoroughbred horses discontinuing a racing career [21].

Objective measurements of the acute stress response may be used to develop more robust phenotypes for the identification of genetic markers associated with differences in the acute stress response. Furthermore, robust phenotypes may make it possible to identify variation within the Thoroughbred population and indicate vulnerability or strengths in adjusting to the rigors of the racing training environment [22]. It has been suggested that genetic screening processes for horses prior to entering the training environment could help identify animals that need to be proactively managed to reduce unnecessary stress for animals that may not be well suited to high performance training environments [23].

The aim of this study was to test the hypothesis that salivary cortisol concentration may be used to objectively detect individual variation in the stress response to being backed by a jockey for the first time. By establishing this, salivary cortisol concentration may be used in further investigations as an objective phenotype for the acute stress response in a training environment. Reliable phenotyping of temperament and acute stress response traits may be useful in the continued efforts to enhance racehorse welfare.

## Materials and methods

### Data collection

**Ethics statement.** The University College Dublin Animal Research Ethics Committee approved the research (AREC-E-17-39-Katz), with informed owner consent (verbal, witnessed by three named authors) obtained from an established research contributor for all procedures.

**Animal cohorts.** The study cohort was comprised of $n = 96$ ($n = 46$ female, $n = 50$ male) yearling Thoroughbred horses born in 2017 that were the progeny of 18 sires (Table 1).

The horses were privately owned, with all management and training decisions made by a single trainer. The horses were housed and trained at the same race-training yard for the study period and had been born and raised at the same stud farm for 13–15 months previously. At the stud farm they were weaned, raised in groups at grass and introduced to simple handling procedures such as being led by halter and veterinary inspection. Between the ages of 13–15 months, horses were transported to a nursery yard prior to commencement of the study. At the nursery yard the young horses were acclimatised to being stabled individually and introduced to exercise on automated horse walkers. At the trainer's discretion, horses, aged 15–18 months, were transported a short distance to the main training yard. The horses had not been prepared for sales and did not have any other major variation in human-mediated interventions or novel experiences at the stud farm or nursery yard that we were aware of.

The cohort of horses used for the time course experiment comprised of $n = 5$ female horses that had been housed in the main training yard for a minimum of three weeks and had completed the early training events including being driven in hand in long reins, being backed by a jockey and being ridden on the gallops with a group of other horses.

**Saliva collection protocol.** Cortisol concentrations were measured from saliva samples collected from horses standing in their own stalls. A dental sponge was held with blunt-edge forceps inserted into the corner of the horse's mouth. The sponge was carefully moved over, under and around the tongue and cheek absorbing the available saliva for ~30s before being placed into a 1.5ml sterile collection tube which contained a small piece of pre-prepared plastic drinking straw in the base to ensure that saliva collected was not reabsorbed by the dental sponge following centrifugation prior to storage [24, 25]. Samples were refrigerated within 30

mins of collection and centrifuged at $1500 \times g$ for 10 mins [9] before the dental sponge and drinking straw were removed and placed into a separate sample tube. The saliva retrieved post-centrifugation was aliquoted into a labelled 1.5ml Eppendorf tube. For all time points other than the time course experiment the horses had not received hard feed for two hours prior to sampling and had not been exercised within the two hours prior to sampling.

## Sampling time course and events

Details for the sampling timepoints provided in Table 1 are expanded below, with an explanation of each timepoint.

**Resting nursery.**   To establish a baseline cortisol measurement for horses prior to movement to the main training yard salivary samples were collected from stabled horses at rest at the nursery yard. Samples were collected between 14:00–15:30. Considering the daily routine at the main yard this period was deemed most suitable for collecting baseline samples across the two locations since this time of day had minimal activity and events taking place which could potentially trigger a behavioural or stress response and elevate cortisol levels.

**Resting main yard.**   To establish a baseline cortisol measurement for horses after arriving at the main training yard salivary samples were collected from stabled horses at rest within three days of being transported from the nursery yard. Samples were collected between 14:00–15:30.

**Follow-up main yard.**   Salivary samples were collected from stabled horses at rest at the main training yard following 2–3 weeks after entering the main training yard. Samples were collected between 14:00–15:30.

**Resting time-course in main yard.**   To investigate whether time-of-day influenced cortisol concentrations salivary samples were collected from $n = 5$ female horses at rest in the main training yard at five two-hour intervals between 08:00–16:00 (T1 –T5). This period encompassed most of the time during which the yard environment and horses were active. Horses received their normal hard feed rations at 06:00 and 12:00 and were exercised on an automated walker for approx. 30–60 mins in both the morning and afternoon. Since exercise is known to influence cortisol concentrations [24] samples were collected at least 1 hour after exercise.

**Pre/post-driving.**   Salivary samples were collected from a stabled horse at rest at the main training yard prior to tack being applied (FD_T1) and ~30 minutes after the horse was 'driven' using long reins (driving) (FD_T2), before the horse had access to drinking water. The time of day of the driving training sessions varied and were between 10:00 and 12:00 and between 15:00 and 16:30.

**Pre/post-backing by a jockey.**   Salivary samples were collected from a stabled horse at rest at the main training yard prior to tack being applied (FR_T1) and ~30 minutes after the first time a horse was backed by a jockey (FR_T2), before the horse had access to drinking water. The time of day of the first backing by a jockey for each horse varied and were between 10:00 and 12:00 and 15:00 and 16:30.

**Pre/post-first time on gallops.**   Salivary samples were collected from a stabled horse at rest at the main training yard prior to tack being applied (FG_T1) and ~30 minutes after the first time the horse was ridden on the gallops with a group of other yearling horses (FG_T2), before the horse had access to drinking water. Sampling took place between 10:00–12:00.

## Cortisol Enzyme-Linked Immunosorbent Assay (ELISA)

Cortisol concentrations were measured using the Salimetrics<sup>TM</sup> Salivary ELISA Kit (Salimetrics<sup>TM</sup>, Newmarket, U.K.), an immunoassay previously validated for use in horses [26]. Samples were prepared and processed according to manufacturer's instructions. Samples were

thawed, mixed and centrifuged before being loaded onto the coated plate. Enzyme conjugate was added, and the plate mixed using a plate rotator. The plate was washed, and substrate solution added to each well. The plate was mixed again using a plate rotator before stop solution was added. The plate was then mixed a final time using a plate rotator before being read. Plates were read using a microplate spectrophotometer (Epoch, BioTek® Instruments Inc.) within 10 mins of adding stop solution at 450nm, with a secondary filter correction at 490-492nm.

### Data analysis

All data formatting and statistical analyses were performed within RStudio [version 3.6.0] [27]. Raw data files were formatted using *Tidyverse* [version 1.2.1] [28]. Cortisol concentrations were calculated using a custom R script following the protocol set out by the Salimetrics™ Salivary ELISA kit.

Two outliers were identified where it appeared that the assay had failed, and the observed values were below values from the lowest resting values detected across the entire cohort. These results were therefore removed from further analysis.

Only paired samples in which both pre- and post-event values were available for the same individual were included in analyses of responses to events. For all other resting cohorts all available samples were included in the analysis.

Quality of experimental data was evaluated using the inter- and intra-assay coefficients of variation (CV). All data was normalised using a log transformation.

In R Studio [version 3.6.0] [29] an ANOVA was performed to test for difference among the five timecourse timepoints (T1-T5, Table 2) sampled. Paired t-tests were performed to test for difference between the five timecourse timepoints. An ANOVA was performed to test for difference between the three resting timepoints: (RN, RY_T1, and RY_T2). Paired t-tests were performed to test for significant difference between each pre / post training event sample (FD_T1 Vs FD_T2, FR_T1 Vs FR_T2, and FG_T1 Vs FG_T2). A test statistic of $P < 0.05$ was considered a significant result in this study. Details of all statistical tests are provided in S2-S6 Tables.

## Results

The inter-assay coefficient of variation (CV) among all samples was 0.96%. The mean intra-assay CV was 0.83% (SD±0.69), and the range in intra-assay CV among the nine plates was 0.29–2.55%. All CVs were calculated from raw read data.

For intra-assay CV values >10% the samples were assayed again on another plate where possible, and the value from the second test was used for downstream analysis. There was a small number of samples ($n = 6$) where insufficient sample volume was available for the repeat assay and the original value was used to retain data for the largest possible cohort size for each event.

**Table 2. Cortisol concentrations of timecourse cohort.**

| Timepoint | Mean (nmol/L) | ± SD | Range (nmol/L) |
|---|---|---|---|
| Timecourse 1 (T1) | 3.33 | 0.63 | 2.55–4.00 |
| Timecourse 2 (T2) | 1.95 | 0.34 | 1.55–2.44 |
| Timecourse 3 (T3) | 2.19 | 0.53 | 1.50–2.76 |
| Timecourse 4 (T4) | 1.94 | 0.48 | 1.54–2.66 |
| Timecourse 5 (T5) | 2.18 | 0.58 | 1.56–2.90 |

Across all samples, cortisol concentrations ranged from 1.03–36.18nmol/L. The cortisol concentrations for all samples at all timepoints are given in S1 Table.

## Timecourse results

Fig 1 presents cortisol concentrations in nmol/L for timecourse samples collected at T1 –T5 for *n* = 5 horses.

Mean cortisol concentrations, ranges and SD for timepoints 1–5 (T1-T5) are given in Table 2.

Cortisol concentrations varied significantly (ANOVA, $F$ = 4.83, df = 4, $P$ < 0.05) among resting samples across five timepoints. Differences were observed in concentrations at timepoint 1 [08:00] (T1, Table 3) (Paired T-test, $P$ < 0.05, Table 3), compared to concentrations at the later timepoints (T2-T5, Table 3). The later four timepoints (T2-T5, Table 3) were not different (Paired T-test, $P$ > 0.05, Table 3).

## Resting results

Fig 2 shows cortisol concentrations in nmol/L for samples collected at three resting timepoints (Rest Nursery [RN], Resting Main Yard [RY_T1], Resting Main Yard follow-up [RY_T2]). There was no significant (ANOVA, $F$ = 2.98, df = 2, $P$ > 0.05) difference in mean cortisol concentration among samples collected at these three resting timepoints. Mean concentrations, ranges and SD are all presented in Table 4.

## Pre / post training event results

Cortisol concentrations for all pre and post training event in nmol/L are visualised in Fig 3.

**First time being driven.** Cortisol concentrations were significantly higher (paired t-test, $t$ = -4.493, df = 5, $P$ < 0.005 (0.006)) for samples collected following first time being driven (FD_T2) as compared to samples collected before the event (FD_T1) (Fig 4). Mean and range values (nmol/L) are detailed in Table 5.

**First time being backed by a jockey.** Cortisol concentrations were significantly higher (paired t-test, $t$ = -10.48, df = 33, $P$ = $4.928 \times 10^{-12}$) for samples collected following first time being backed (FR_T2) compared to samples collected before the event (FR_T1) (Fig 5). Mean and range values (nmol/L) are detailed in Table 5.

**First time on the gallops.** Cortisol concentrations were significantly higher (paired t-test, $t$ = -11.22, df = 9, $P$ = $1.364 \times 10^{-6}$) for samples collected following the first time on the gallops (FG_T2) as compared to samples collected before the event (FG_T1) (Fig 6). Mean and range values (nmol/L) are detailed in Table 5.

## Discussion

We measured variation in salivary cortisol concentrations in Thoroughbred yearlings entering training and experiencing key early training events not previously encountered. As variation in cortisol concentration due to normal circadian hormonal fluctuations has been previously reported [10] we first performed a timecourse experiment sampling a representative cohort of horses every two hours over an 8-hour period during which normal yard functions were performed. All sampling was performed in the horses' own stables during periods of rest. The mean cortisol concentration for the first timepoint (08:00) was significantly higher than all other timepoints (10:00–16:00), an observation concordant with previously reported findings of a diurnal cortisol rhythm that peaks in the morning during the waking phase [10, 30, 31]. There was no significant difference in mean cortisol levels across the other time points.

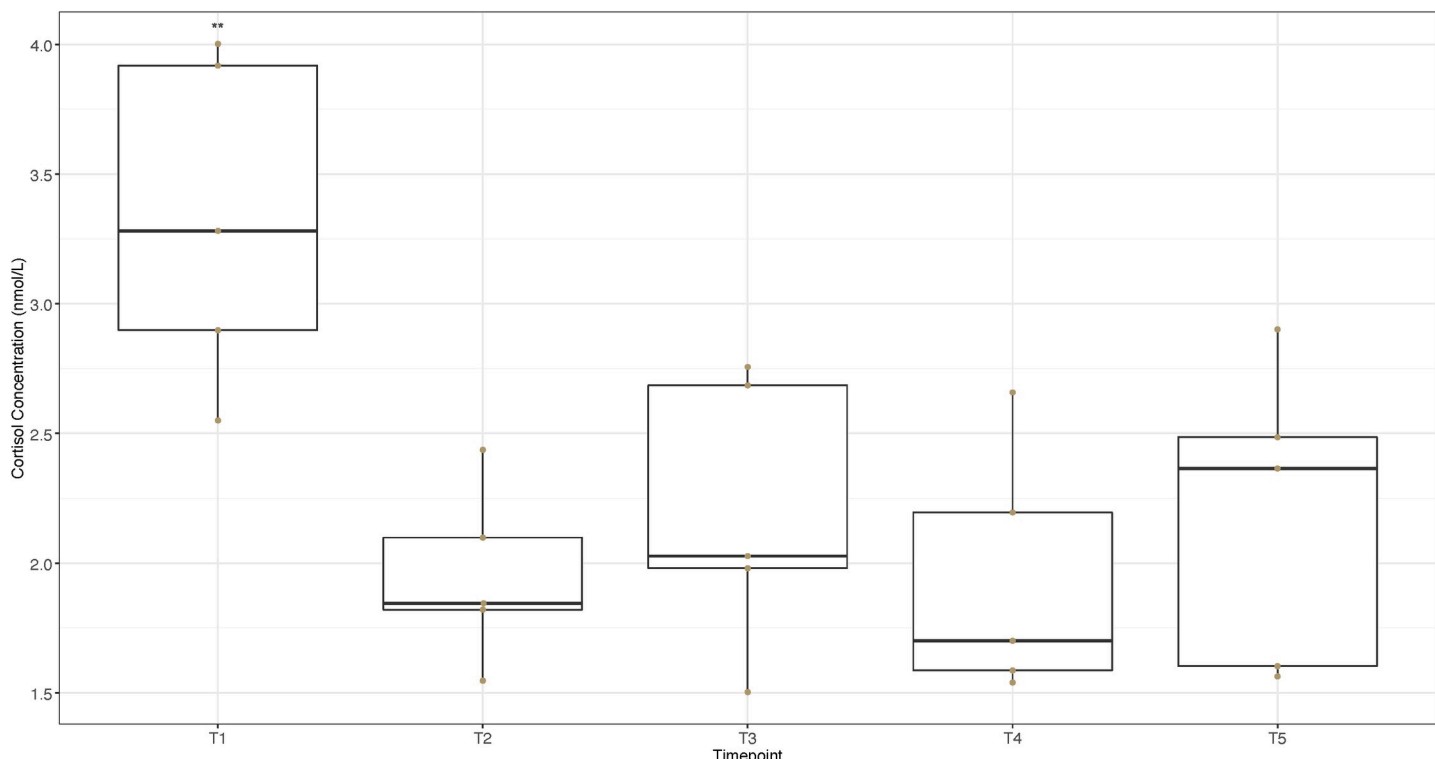

**Fig 1. Salivary concentrations of the stress hormone cortisol at over an eight-hour period.** The boxplots indicate the median and interquartile range of salivary cortisol concentrations in nmol/L sampled every two hours from (n = 5) horses at rest between 08:00 and 16:00 (inclusive). Error bars represent the 5th and 95th percentiles while the dots represent the individual cortisol values for the horses in each group. Paired T-test; ** = P <0.05.

Considering this and the normal daily routine of the horses, experimental 'resting' samples were collected between 14:00–15:30, since there was minimal activity during that period, thus reducing the potential to measure behavioural/stress response to environmental stimuli. Since the horses included in the study were in early training for racing it was not possible to adjust the schedule of the other training events to this period only. However, all experimental sampling was conducted between 10:00–16:00 to reduce any variation resulting from the factors discussed.

Moving from a familiar environment to a new environment may have the potential to induce a stress response for a naïve horse [32] and therefore before the horses moved from the nursery yard to the main training yard baseline cortisol measurements were collected. Follow up samples were collected between 24- and 72-hours following arrival in the main yard and 2–3 weeks after entering the new training environment. No significant difference in mean cortisol concentrations was observed across these three timepoints. Road transport [32] and novelty presented to naïve animals [12, 33] have both been previously investigated in horses using

**Table 3. *P*-value results from paired T-tests showing differences between individual timepoints from timecourse experiment (*n* = 5).**

|  | Timecourse 1 08:00 | Timecourse 2 10:00 | Timecourse 3 12:00 | Timecourse 4 14:00 |
|---|---|---|---|---|
| **Timecourse 2 10:00** | 0.0018 | - | - | - |
| **Timecourse 3 12:00** | 0.0167 | 0.3748 | - | - |
| **Timecourse 4 14:00** | 0.0252 | 0.9028 | 0.4654 | - |
| **Timecourse 5 16:00** | 0.0059 | 0.366 | 0.9113 | 0.4865 |

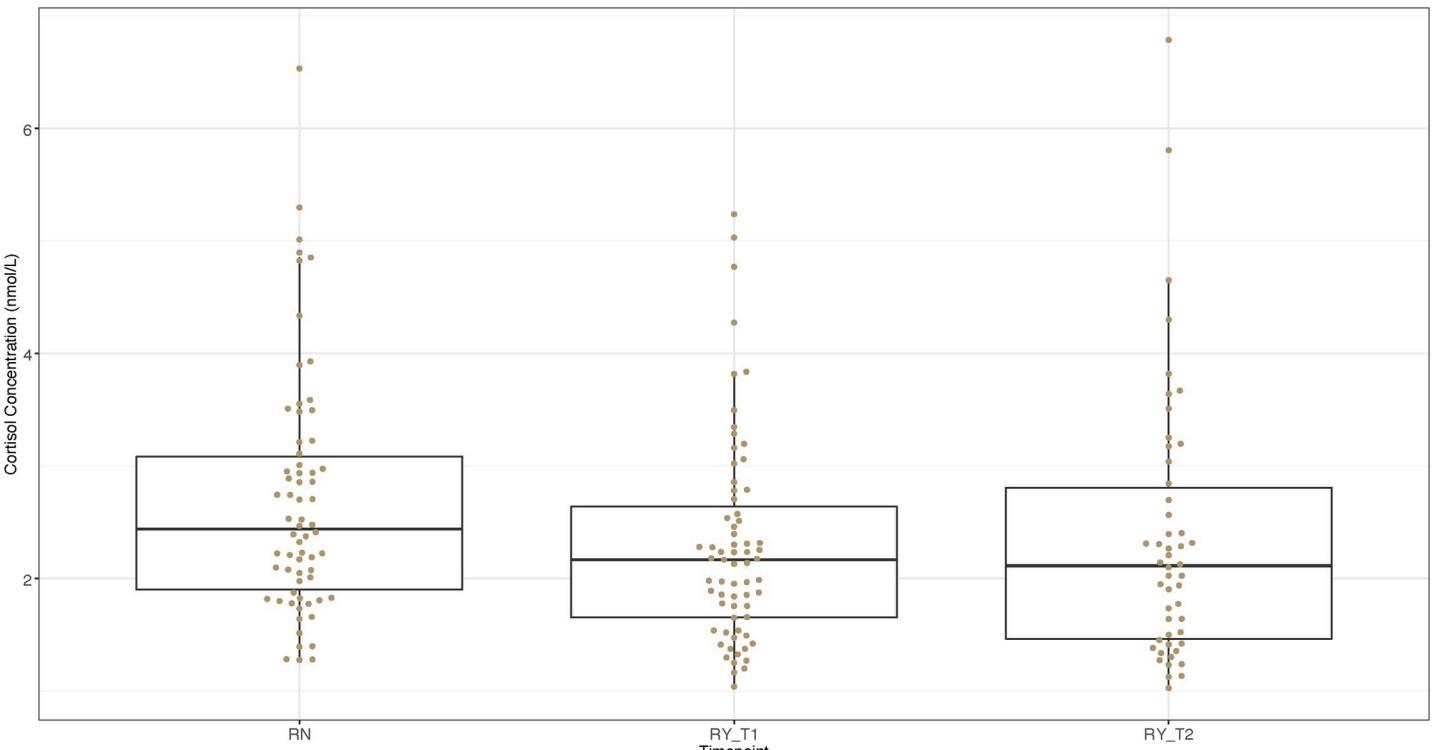

**Fig 2. Salivary concentrations of the stress hormone cortisol among horses at rest when located at the nursery yard and after being moved to the main training yard.** The boxplots indicate the median and interquartile range of salivary cortisol concentrations in nmol/L sampled from horses at rest located at the nursery yard (RN; n = 66), within three days of entering the main training facility (RY_T1; n = 67), and after two-three weeks of being housed at the main training facility (RY_T2; n = 50). Error bars represent the 5th and 95th percentiles while the dots represent the individual cortisol values for the horses in each group. No significant differences detected (ANOVA $F$ = 2.98, df = 2, $P$>0.05).

salivary cortisol, and were found to provoke an acute stress response. However, here the purpose was not to measure the acute stress response to the transport process itself, but rather to investigate potential lasting effects on baseline cortisol levels indicating insufficient adjustment and increased stress in response to the novel environment. Our results indicate that the new environment encountered by the yearlings in the training yard did not appear to cause a prolonged stress response. It is unlikely that movement from the nursery yard to the main training yard had no effect on cortisol levels; however, the response to transport may be acute, which would not have been detected at the time point measured here which allowed for a period of adjustment. This observation may reflect the considerable behavioural plasticity of young animals [34] enabling adaptation and learning.

Here, the greatest variation in cortisol concentration levels were observed in the post-event samples of the novel early training events. These early training events: the first time being driven using long-reins (3.88–19.62 nmol/L ± 5.66), the first time being backed by a jockey (2.71–36.18 nmol/L ± 7.21) and the first experience of being ridden out in a group on an

**Table 4. Cortisol concentrations of resting cohorts.**

| Timepoint | Mean (nmol/L) | ± SD | Range (nmol/L) |
|---|---|---|---|
| Resting Nursery (RN) | 2.69 | 1.08 | 1.28–6.53 |
| Resting Nursery Main Yard (RY_T1) | 2.30 | 0.92 | 1.04–5.24 |
| Follow-up resting Main Yard (RY_T2) | 2.36 | 1.19 | 1.19–6.79 |

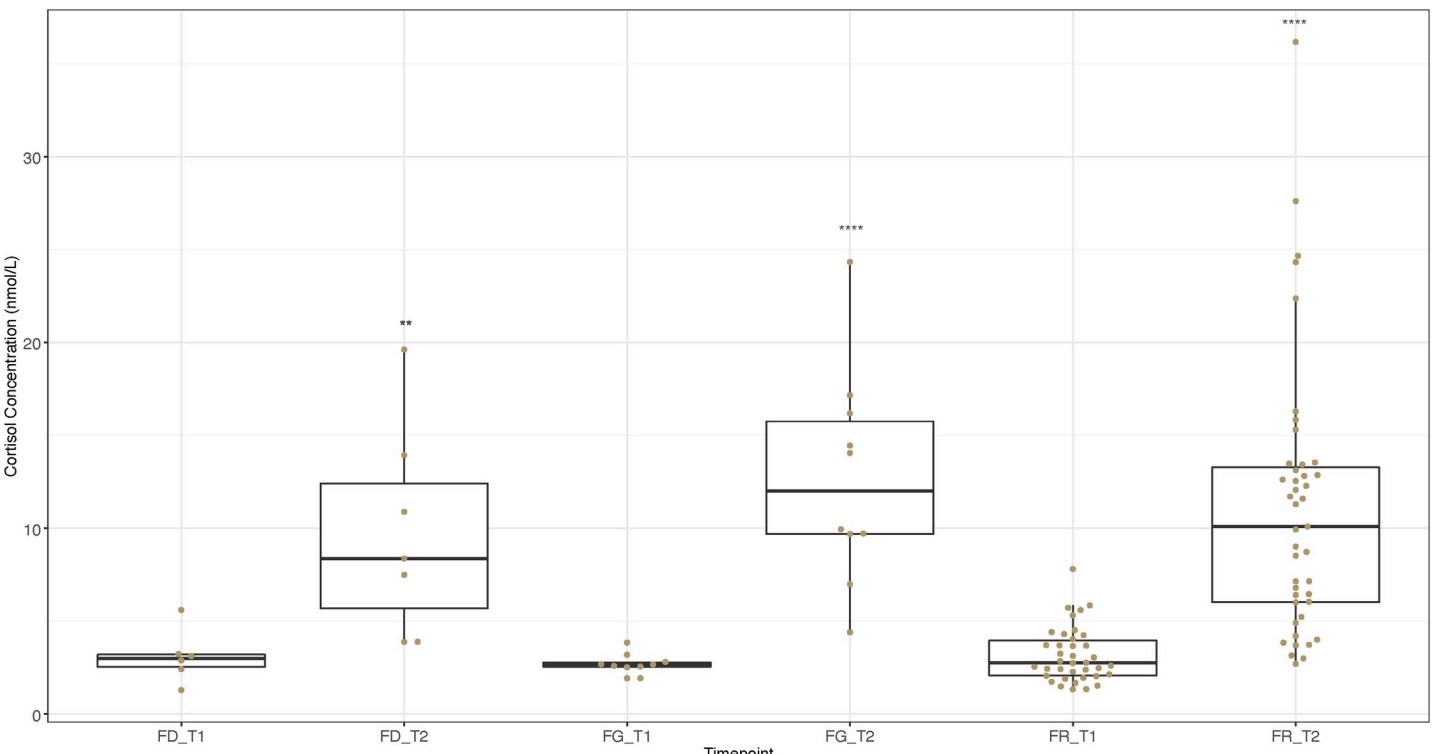

**Fig 3. Change in the salivary concentrations of the stress hormone cortisol between pre- and post-training for milestone training events.** The boxplots indicate the median and interquartile range of salivary cortisol concentrations in nmol/L pre- (T1) and post-training (T2) for first time being driven (FD; n = 6), first time being backed (FR; n = 34) and first time on the gallops (FG; n = 10). Error bars represent the 5th and 95th percentiles while the dots represent the individual cortisol values for the horses in each group. Paired t-test; ** = P<0.01, **** = P<0.0001.

exercise track (the gallops) (4.40–24.33 nmol/L ± 5.77) are all part of the initial training that a young horse undergoes prior to exercise conditioning and competitive racing. Novel training events, particularly first time being backed by a jockey, has previously been shown to be highly stressful for naïve horses [12]. Our results are in keeping with previous research highlighting first backing as a key training milestone eliciting an acute stress response during the early training period. Furthermore, the results presented here highlight individual variation in response to first backing. This variation in salivary cortisol levels is indicative of individual differences in HPA responsivity.

Handler body language and emotional state are known to have a significant effects on anticipatory behaviour [35] as well as other stress response measures such as heart rate/heart rate variability [36] suggesting an influence on the emotional reactivity of the horse. The increase in salivary cortisol concentration in this dataset cannot exclude the influence of handler-induced variation as a result of varying body language [37], emotional state [35] or skill [38]. However, a comparison of two first-saddling and two first-backing methods showed that regardless of handler style and method, the event of first backing by a jockey was always most stressful to a naïve horse [15]. The training style used by all handlers who worked with the horses in this study were more similar than the two distinctly different styles investigated by Kędzierski *et al.* [15]. Therefore, while some variation may be a result of different handlers, the significant variation observed is likely to be related to other factors. One such factor that may influence the variation observed is the horses' temperaments.

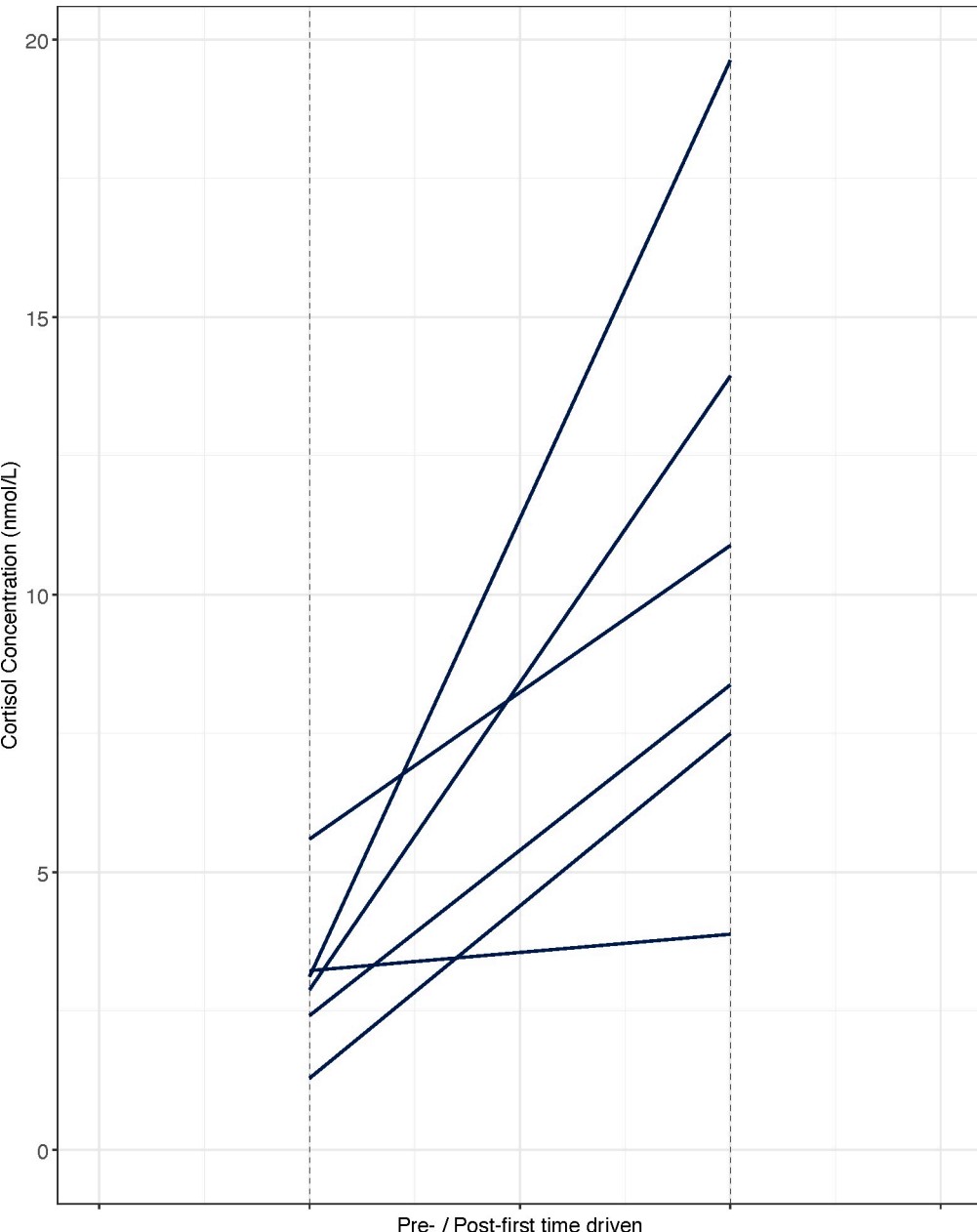

**Fig 4. Cortisol concentrations (nmol/L) pre / post first time being driven for** *n* **= 6 Thoroughbred yearlings.** Each line represents a horse's salivary cortisol response to the training event.

Another factor that may influence the observed variation is age / naivety of the horses studied. Interestingly, a recently published study showed that experienced, non-Thoroughbred, horses had overall lower salivary cortisol levels than naïve horses [39]. This study found no significant increase in salivary cortisol level in response to training events similar to those described by this research. However, it is difficult to make a direct comparison between the two experiments since the previously reported study measured salivary cortisol in two different breeds of horses. Indeed, breed could be another factor influencing the observed differences. It

**Table 5. Cortisol concentrations for pre / post training event timepoints.**

| Timepoint | Mean (nmol/L) | ± SD | Range (nmol/L) |
|---|---|---|---|
| Pre-driving (FD_T1) | 3.09 | 1.42 | 1.29–5.60 |
| Post-Driving (FD_T2) | 9.73 | 5.66 | 3.88–19.62 |
| Pre-backing (FR_T1) | 3.17 | 1.48 | 1.33–7.81 |
| Post-backing (FR_T2) | 11.09 | 7.21 | 2.71–36.18 |
| Pre-first time on gallops (FG_T1) | 2.67 | 0.56 | 1.93–3.85 |
| Post-first time on gallops (FG_T2) | 12.69 | 5.77 | 4.40–24.33 |

is noteworthy therefore that the Thoroughbred is a breed which has been under selection for athletic traits for >300 years which may include reactivity to stimuli.

Variation observed across all pre-event timepoints, collected from the horses when stabled at rest and prior to any training activities, was consistently significantly lower than those collected post-event. This contrast in the significant increase in cortisol concentration post-event might reflect the response to novelty as already discussed, but it may also allude to a number of additional influencing factors. One such factor could be a variation between individuals as to when salivary cortisol concentrations peak. Early training events such as being driven (using long reins) and being mounted by a jockey for the first time are experiences which involve a horse being required to perform a task away from their familiar housing and stable neighbours. Although horses are housed in individual looseboxes they can still visually and audibly communicate with nearby companions which has been shown to reduce stress when compared with completely isolated housing [40]. Since horses are herd animals with highly complex social hierarchies in the feral setting [41] being segregated and asked to carry out novel tasks such as learning the riders aids during early groundwork may be a cause of considerable psychological stress. Recent research has shown horses are capable of reading emotion [42] and remembering positive and negative experiences [43]. As already discussed, while handler style has not been shown to have significant impact on cortisol response to first backing [15] it is possible however, that environmental observations of other horses undergoing training nearby may have an impact on their response [42].

Detecting individual levels of stress response is useful in the management of animals, particularly those that are considered high-performing such as racehorses [16]. Racehorse trainers commonly describe Thoroughbred horses as being sensitive and responsive and it is possible that these subjective observations are underpinned by differences in stress reactivity. While stress has potential to increase exercise capacity [13], it is likely it also has longer-term welfare implications [44, 45]. Behavioural response and stress tolerance within the training environment are thought to have a familial basis and 'tractability' measured by subjective assessment has been demonstrated as having a genetic component in the Thoroughbred [46].

The data collection access within the racing yard was not controlled by the researchers, and as such the timing of the training sessions and horse-handler distributions were determined by the yard staff. However, having access to racehorses in training at a single racing yard as opposed to a cohort situated across several yards was beneficial to reduce variation for many other variables which often pose a challenge to behavioural research. For example, husbandry practice, specifically type of housing, is known to provoke a stress response [30]. All horses were stabled, and subject to uniform feeding and husbandry practices throughout the period of study, which is a major strength of the experimental set-up at the host yard, especially since dietary carbohydrates are known to have an effect on circulating cortisol [47]. Variation in cortisol levels have also been previously reported to be associated with experience and training of athletes [31] but since all yearlings were entering training their experience can be considered

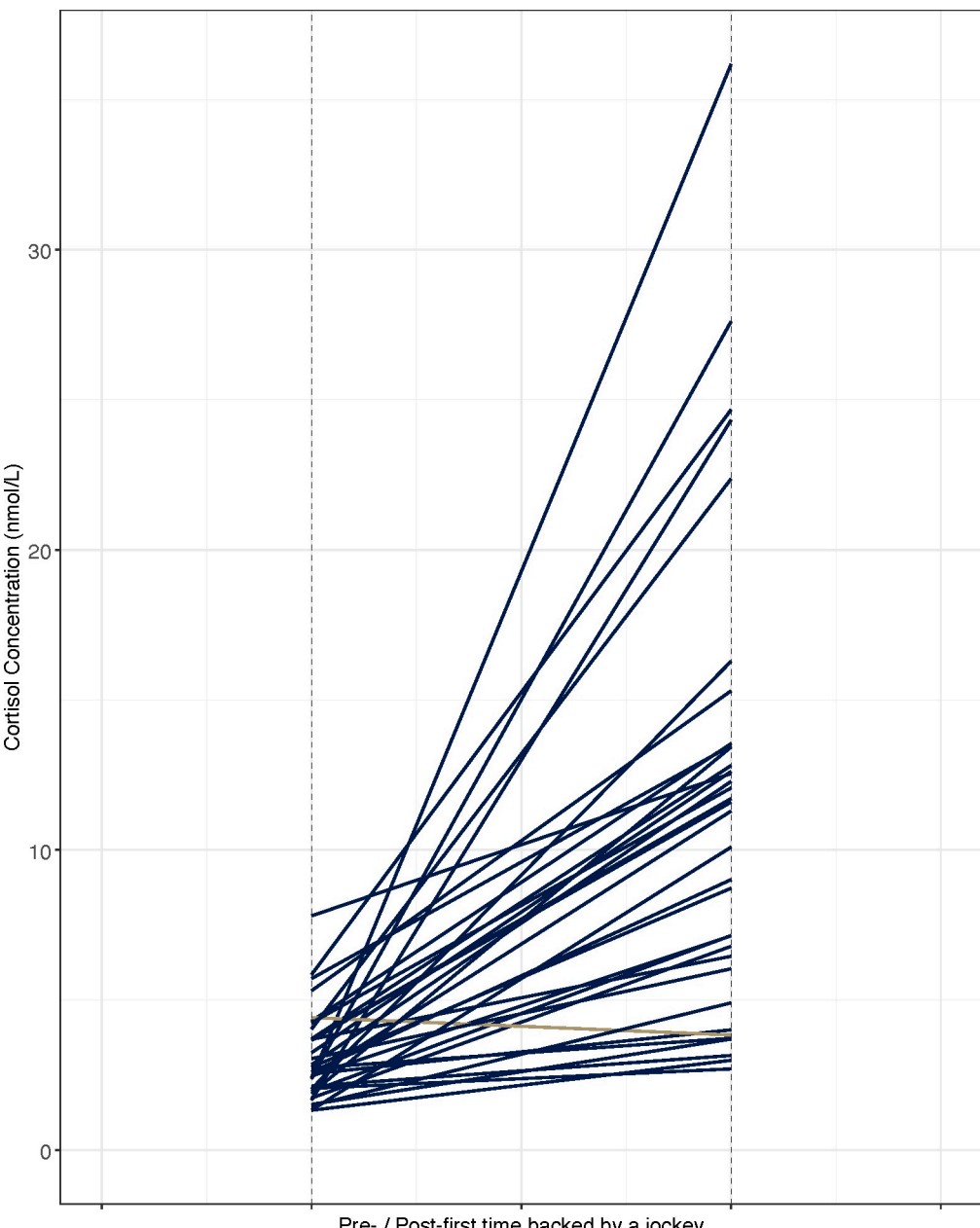

**Fig 5. Cortisol concentrations (nmol/L) pre / post first-backing by rider for *n* = 34 Thoroughbred yearlings.** Each line represents a horse's salivary cortisol response to the training event.

equivalent. Finally, where previously, resting cortisol variation has been associated with the time of year [48, 49], it is unlikely the variations in salivary cortisol concentrations measured here are a result of seasonal effects as sampling took place over a three-month period between October and January (2018–19).

Unavoidable limitations to this work, such as the variation in time when resting samples (RY_T2) were collected at the main training yard following initial training, reflects what is already known in the racing industry, that trainers respond and adjust the regimen in response to their interpretation of the behavioural feedback of individual horses, with some horses

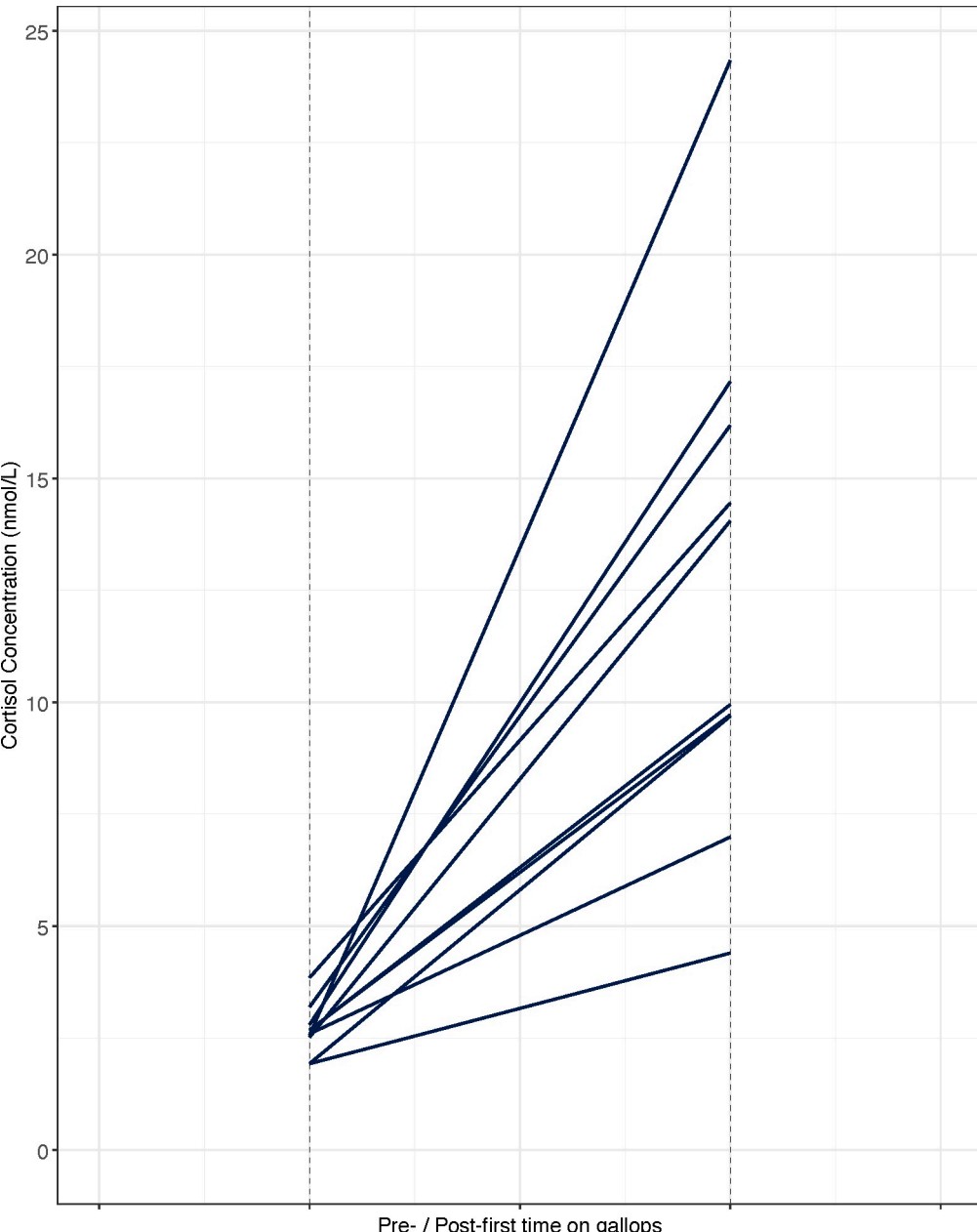

**Fig 6. Cortisol concentrations (nmol/L) pre / post first time on gallops for *n* = 10 Thoroughbred yearlings.** Each line represents a horse's salivary cortisol response to the training event.

taking longer than others to complete early training events. Since the nature of this data collection was a convenience sample, the timecourse experiment was performed using only female horses (detailed in Tables 2 and 3) as these horses were available. The methodological, technical and operational nature of these limitations is not unique to equine or behavioural research, and such constraints are commonly presented when working with real-world datasets [50].

## Conclusions

The purpose of this study was to determine if salivary cortisol is a reliable method for objectively detecting individual variation in the stress response of horses entering pre-race training. A significant difference between cortisol concentrations measured in salivary samples taken before and after encountering early training events, as well as individual variation observed at these three novel training events, supports the prior hypothesis that salivary cortisol can be used as a biomarker of the neurobiological stress response. This biomarker is intended for use with temperament / behaviour measures for further analyses in future studies. The wide range in cortisol response between horses before and after first-backing by a jockey showed individual variation and indicates that first backing would be suitable to provide an objective phenotype for future research. Being able to effectively phenotype horses for stress response may enable further study into underlying mechanisms, such as genotype, influencing stress in the Thoroughbred horse. At a time when welfare of horses in sport is under the spotlight this research provides scientific basis for better understanding stress in the racehorse.

## Supporting information

**S1 Table. Table of all salivary cortisol concentrations collected across all timepoints.**
(DOCX)

**S2 Table. Table of statistical details for ANOVA for timecourse samples.**
(DOCX)

**S3 Table. Table of P values for paired t-tests for timecourse samples (df = 4).**
(DOCX)

**S4 Table. Table of T values for paired t-tests for timecourse samples (df = 4).**
(DOCX)

**S5 Table. Table of statistical details for ANOVA for resting samples.**
(DOCX)

**S6 Table. Table of statistical details for milestone training event samples.**
(DOCX)

## Acknowledgments

The authors wish to thank J.S. Bolger for access to his horses, as well as staff at both Glebe House and Beechy Park stables for their assistance with this research. Thanks also to Dr. Michael Griffin and Ms. Jess Stallard for their help piloting saliva sample collection. This research was funded by Plusvital Ltd.

## Author Contributions

**Conceptualization:** Amy R. Holtby, Lisa M. Katz, Keith J. Murphy, Emmeline W. Hill.

**Data curation:** Amy R. Holtby, Beatrice A. McGivney.

**Formal analysis:** Amy R. Holtby, Beatrice A. McGivney.

**Funding acquisition:** Emmeline W. Hill.

**Investigation:** Amy R. Holtby.

**Methodology:** Amy R. Holtby, Lisa M. Katz, Emmeline W. Hill.

**Project administration:** Amy R. Holtby, Emmeline W. Hill.

**Resources:** John A. Browne, Lisa M. Katz.

**Software:** Amy R. Holtby, Beatrice A. McGivney.

**Supervision:** Beatrice A. McGivney, John A. Browne, Emmeline W. Hill.

**Visualization:** Amy R. Holtby.

**Writing – original draft:** Amy R. Holtby, Emmeline W. Hill.

**Writing – review & editing:** Beatrice A. McGivney, John A. Browne, Lisa M. Katz, Keith J. Murphy, Emmeline W. Hill.

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
