## [Decision Letter · Decision Letter 0]

14 Dec 2022

PONE-D-22-23041Variation in salivary cortisol responses in yearling Thoroughbred racehorses during their first year of trainingPLOS ONE

Dear Dr. Holtby,

Thank you for submitting your manuscript to PLOS ONE. After careful consideration, we feel that it has merit but does not fully meet PLOS ONE’s publication criteria as it currently stands. Therefore, we invite you to submit a revised version of the manuscript that addresses the points raised during the review process.

Thank you for the submission of the manuscript. The reviewers have requested some edits to the manuscript. Could you please address these and resubmit the manuscript with a cover letter addressing the reviewers comments.

We look forward to receiving your revised manuscript.

Kind regards,

Chris Rogers

Academic Editor

PLOS ONE

Journal Requirements:

“This research was carried out with the financial support of Plusvital Ltd. Plusvital is an equine nutrition and genetic testing company in which EWH is a shareholder.”

4. Thank you for providing the following Funding Statement: 

“ARH and BMcG are employees of Plusvital Ltd. EWH is Chief Scientific Officer for Plusvital Ltd.”

We note that one or more of the authors is affiliated with the funding organization, indicating the funder may have had some role in the design, data collection, analysis or preparation of your manuscript for publication; in other words, the funder played an indirect role through the participation of the co-authors.

If the funding organization did not play a role in the study design, data collection and analysis, decision to publish, or preparation of the manuscript and only provided financial support in the form of authors' salaries and/or research materials, please review your statements relating to the author contributions, and ensure you have specifically and accurately indicated the role(s) that these authors had in your study in the Author Contributions section of the online submission form. Please make any necessary amendments directly within this section of the online submission form.  Please also update your Funding Statement to include the following statement: “The funder provided support in the form of salaries for authors [insert relevant initials], but did not have any additional role in the study design, data collection and analysis, decision to publish, or preparation of the manuscript. The specific roles of these authors are articulated in the ‘author contributions’ section.”

If the funding organization did have an additional role, please state and explain that role within your Funding Statement.

Please also provide an updated Competing Interests Statement declaring this commercial affiliation along with any other relevant declarations relating to employment, consultancy, patents, products in development, or marketed products, etc. 

Reviewers' comments:

Reviewer's Responses to Questions

**Comments to the Author**

1. Is the manuscript technically sound, and do the data support the conclusions?

Reviewer #1: Partly

Reviewer #2: Yes

2. Has the statistical analysis been performed appropriately and rigorously? 

Reviewer #1: Yes

Reviewer #2: Yes

3. Have the authors made all data underlying the findings in their manuscript fully available?

Reviewer #1: Yes

Reviewer #2: Yes

4. Is the manuscript presented in an intelligible fashion and written in standard English?

Reviewer #1: Yes

Reviewer #2: Yes

5. Review Comments to the Author

Reviewer #1: This is a well-described and easy-to-follow paper in which all the experimental procedures were correctly described, the statistical analysis was appropriate (although with some concerns), and the discussion was correctly developed.

However, quite comments have been made by this reviewer related, mainly to the discussion results.

Please answer the following questions in detail:

ABSTRACT:

Lines 23-24: Response to Acute stress is not solely produced by the HPA but also by the autonomic nervous via the sympathetic system and Sympathetic adrenal medullary, which are faster in acting after an acute stressor and are responsible for that fight-or-flight response.

Lines 29-34: This is not the study's main objective. Therefore, it could be better to first describe the results from the other experiment (evaluate the acute stress by the pre- and post-novel training events) and then describe this. Since this first experiment only was performed, if this reviewer understood correctly, to check what would be the basal salivary cortisol levels at resting and if daily variations or other external environment variations could bias the salivary cortisol results for the main experimental activity of this study: if variations in horses' salivary cortisol can explain horses' variations due to their temperaments after backing by a jockey for the first time.

Line 44 (Keywords): This reviewer advises including the word "saliva".

METHODS:

Line 161: When authors explain horses did not receive hard feed, did it mean neither hay for two hours? This reviewer thinks to avoid feed contamination in the saliva sample recollections, isn't it? Salivary contamination by feed can influence salivary cortisol concentrations [1].

Lines 232-233: If distribution data behaved as non-parametric distribution, ANOVA and t-test would not be the better to perform the statistical tests unless the authors previously transformed the data to make them normal.

DISCUSSION:

Lines 348-356: Although not the main objective of this study, this pre-experiment was performed in females, not in males or geldings. However, it is well known that this daily cortisol behaviour is in both sexes. However, this must be figurated in the limitation section.

Lines 397-398: This could be better if, during this study, the authors had measured the behaviour to evaluate the different temperaments of each horse and linked it with the cortisol concentrations. This must be pointed out in the discussion as a possible cause of cortisol variability after an acute stressor.

CONCLUSION:

Lines 451-454: This is not a piece of new evidence (the results from the present study are not a novelty in this aspect). Moreover, of course, the authors have observed individual variations in cortisol concentrations after acute stress. However, to objectively demonstrate the individual variation in the stress response of horses by measuring salivary cortisol and, therefore, try to use salivary cortisol to "identify variation within the Thoroughbred population and indicate vulnerability or strengths in adjusting to the rigours of the racing training environment"; these results must be linked with behaviour response from them and, therefore, with their temperament.

REFERENCES

1. Contreras-Aguilar MD, Luisa M, Escribano D, Lamy E, Tecles F, Cerón JJ. Effect of food contamination and collection material in the measurement of biomarkers in saliva of horses. Res Vet Sci. 2020;129: 90–95. doi:10.1016/j.rvsc.2020.01.006

Reviewer #2: Congratulations for a nice study on young horses' stress response during their early learning experiences. The study is sound, even if in some places has a limited sample size; the methods regarding salivary sampling or hormone assay are fine, and the results are reported clearly.

I have only a few minor comments, mainly editorial.

- l.393 ff. In addition to the cited papers, there's a new one out there that reports no elevated stress response to early training. Although their paper examines training tasks that have limited overlap with yours, it might be worth considering. Their finding is basically that early training does not elevate cortisol, except when training is physically demanding enough to cause the elevation. Niittynen et al, 2022 App. Anim. Behav. Sci.

- l. 271, 295, 310 and elsewhere: where you report the p-values, please also always report the test statistic and df.

- Figure legend 3. The text indicates that there's significance indicators in the figure, but there are none.

- I think some of the tables / figures are somewhat unnecessary. I would leave out altogether Table 3 and Figure 2, and move to the Supplement Figures 4, 5 and 6.

6. PLOS authors have the option to publish the peer review history of their article (what does this mean?). If published, this will include your full peer review and any attached files.

Reviewer #1: No

Reviewer #2: **Yes: **Sonja Koski

---

## [Author Response · Author response to Decision Letter 0]

2 Mar 2023

Dear editors / reviewers,

Many thanks for the consideration of our manuscript. All questions and comments have been responded fully to in the file named "Response to Reviewers" attached to the submission.

Kind regards,

---

## [Editor Report · Decision Letter 1]

23 Mar 2023

Variation in salivary cortisol responses in yearling Thoroughbred racehorses during their first year of training

PONE-D-22-23041R1

Dear Dr. Holtby,

We’re pleased to inform you that your manuscript has been judged scientifically suitable for publication and will be formally accepted for publication once it meets all outstanding technical requirements.

Kind regards,

Chris Rogers

Academic Editor

PLOS ONE
---

## [Editor Report · Acceptance letter]

29 Mar 2023

PONE-D-22-23041R1 

Variation in salivary cortisol responses in yearling Thoroughbred racehorses during their first year of training 

Dear Dr. Holtby:

I'm pleased to inform you that your manuscript has been deemed suitable for publication in PLOS ONE. Congratulations! Your manuscript is now with our production department. 

Kind regards, 

on behalf of

Dr. Chris Rogers 

Academic Editor

PLOS ONE